# Rapid Nondestructive Simultaneous Detection for Physicochemical Properties of Different Types of Sheep Meat Cut Using Portable Vis/NIR Reflectance Spectroscopy System

**DOI:** 10.3390/foods10091975

**Published:** 2021-08-24

**Authors:** Yanlei Li, Xiaochun Zheng, Dequan Zhang, Xin Li, Fei Fang, Li Chen

**Affiliations:** Chinese Academy of Agricultural Sciences, National Risk Assesment Laboratory of Agro-Products Processing Quality and Safety, Institute of Food Science and Technology, Ministry of Agriculture and Rural Affairs, Beijing 100193, China; liyanlei2021@163.com (Y.L.); zhengxiaochun@caas.cn (X.Z.); zhangdequan@caas.cn (D.Z.); lixin@caas.cn (X.L.); fangfei@caas.cn (F.F.)

**Keywords:** rapid detection, meat quality, different cut types, sheep, Vis/NIR

## Abstract

The visible and near-infrared spectroscopy (Vis/NIRS) models for sheep meat quality evaluation using only one type of meat cut are not suitable for other types. In this study, a novel portable Vis/NIRS system was used to simultaneously detect physicochemical properties (pH, color *L**, *a**, *b**, cooking loss, and shear force) for different types of sheep meat cut, including silverside, back strap, oyster, fillet, thick flank, and tenderloin cuts. The results show that the predictive abilities for all parameters could be effectively improved by spectral preprocessing. The coefficient of determination (*R*_p_^2^) and residual predictive deviation (RPD) of the optimal prediction models for pH, *L**, *a**, *b**, cooking loss, and shear force were 0.79 and 3.50, 0.78 and 2.28, 0.68 and 2.46, 0.75 and 2.62, 0.77 and 2.19, and 0.83 and 2.81, respectively. The findings demonstrate that Vis/NIR spectroscopy is a useful tool for predicting the physicochemical properties of different types of meat cut.

## 1. Introduction

Meat is popular with consumers all over the world because it can provide abundant amounts of valuable nutrients, such as protein, fat, mineral, vitamins, and so on [1,2,3]. Sheep meat is famous for being more tender and having less fat and cholesterol. The demand for sheep meat is strong in many places around the world, and especially in Asia and Australia [4]. With the improvement in living standards and the expansion of international cooperation, the demand for high-quality sheep meat is increasing and its quality has received unprecedented attention [5].

Traditional methods for detecting meat quality attributes include pH meter measurement, the cooking loss method, and the shear force method, but their disadvantages are their destructiveness, sample contamination, and complex sample preparation procedures. As a result, these traditional methods cannot promote the rapid development of the meat industry [6,7]. Therefore, it is important to establish a rapid, non-destructive, and simultaneous meat quality detection. In recent years, Raman spectroscopy and visible and near-infrared (Vis/NIR) spectroscopy, as rapid and non-destructive detection technologies, have been applied for determining various meat quality attributes. However, the large amount and complexity of data obtained by Raman spectrometers and their high cost limit their application to meat quality detection [8,9]. Vis/NIR spectroscopy has significant advantages in terms of acceptable cost and multi-parameter simultaneous fast detection, making it a promising tool for online meat quality detection applications [10,11]. The effective Vis/NIR spectroscopy information, produced in combination bonds and overtone vibrations of molecular groupings including C‒H, O‒H, N‒H, and S‒H, is used for analyzing the features and structures of specific chemical substances [12,13]. As an advanced and efficient method, Vis/NIR spectroscopy has been applied for assessing different meat quality attributes by predicting chemical composition and quality parameters [14,15]. The detection of the chemical composition of meat by Vis/NIR spectroscopy was studied using pork, beef, lamb, and chicken [16,17,18]. The measurements of color (L*, a*, and b*), pH, water-holding capability (WHC), shear force (tenderness), meat spoilage detection, and meat quality grading and classification based on Vis/NIR spectroscopy were previously reported [10,19,20]. However, the prediction models reported in most of the studies were established using only one cut type, resulting in poor practical applicability to other cut types. Different cuts of the carcass have different tissues composition; for instance, there are obvious differences in tenderness and color between the meat parts with more connective tissue and those with less, as well as nutrients, such as protein, fat, and carbohydrates [21]. Fowler et al. [22] explored the potential for the prediction of the intramuscular fat (IMF) content of lamb loin and topside using NIR spectra collected on the topside in situ and concluded that further research was needed to develop models for industry application based on more appropriate calibration data. For most of the meat processors, after sheep carcass segmentation, all meat cuts are quickly obtained and transported without distinguishing their type on the conveyor belt for further processing and finally, they are packaged for commercial sale. Thus, one single model for different types of cuts will reduce costs (labor and spectrometers) and detection time, which will be beneficial for ease of use, especially when used in the cutting room. It is unrealistic for meat processors to apply individual models for each meat cut on the conveyor line in practice. Therefore, establishing a model suitable for multiple cut types would be useful for meat quality detection due to its high usability and detection speed.

To achieve the simultaneous detection of multiple parameters for meat quality, in this study, the pH, color (*L**, *a**, *b**), cooking loss, and shear force of different sheep meat types were predicted as quality indices. Meat color is closely related to the generation of biogenic amine and is regarded as an important quality for determining the purchase intention of consumers [23]. Tenderness and water holding capacity (WHC) are two other important physicochemical indexes of meat quality and are closely related to each other. Both are determined by the physicochemical form and chemical composition of meat. The WHC is usually measured by cooking loss, which affects the color, flavor, and tenderness of the meat. A decrease in the WHC leads to decreases in meat quality, shortening of the storage period, and increases in economic loss, so the detection of WHC is vital to the meat industry [24].

To achieve rapid, nondestructive, and simultaneous online detection, we adopted Vis/NIR spectroscopy to predict the physicochemical properties (pH, *L**, *a**, *b**, cooking loss, and shear force) of different types of commercial sheep meat cuts. Our objectives were: (1) to develop portable Vis/NIR spectroscopy equipment for collecting spectral data; (2) to apply different spectral data pretreatment methods for improving the robustness of the prediction models of six cut types; and (3) to establish prediction models using the partial least square regression (PLSR) method based on two sample set divisions for the simultaneous detection of multiple physicochemical properties for different types of meat cuts.

## 2. Materials and Methods

### 2.1. Preparation of Meat Samples

According to the standard of cutting technical specifications for sheep meat (NY/T 1564-2007), the meat samples used in this experiment were collected from six types of meat cut (silverside, back strap, oyster, fillet, thick flank, and tenderloin cuts) of small-tailed Han sheep carcass in an abattoir. Small-tailed Han sheep is a famous sheep breed in China for its tender and fragrant meat and high percentage of clean meat and is known as the world’s “super sheep”, which is very popular with consumers. To expand the range of sample data and ensure the applicability of the models, the representative carcasses with different body sizes were selected for wide compositional variability. Every day, five carcasses were segmented by operators after chilling at 4 °C for 24 h, then the 30 samples (6 cut types × 5 carcasses) were collected and taken for Vis/NIR spectral data acquisition and quality parameter determination on the same day. The whole sample collection lasted 5 days with a total of 150 samples. Before the acquisition of spectral data, the samples were packed in self-sealed bags and maintained at 4 °C in a refrigerator to maintain stable color and pH. As the spectral information of connective tissue and fat are obviously distinct from that of lean meat, the external and visible fat and connective tissue of each sample were removed using a scalpel to eliminate their adverse effects on modeling ability. The meat samples with larger irregular shapes were further smoothed by cutting the surface to avoid the influence of irregular shapes on spectral information. All samples except the tenderloin cut were trimmed to about 5 cm × 5 cm × 3 cm. Because of the special shape being difficult to cut into cubes, about 100 g of the tenderloin cut was taken as the tested sample. After collecting and measuring the spectra, pH, and color, the sample was immediately placed in a refrigerator maintained at 4 °C until all samples were measured. The samples were then removed to measure the cooking loss and shear force. The specific procedure of each measurement is introduced in the following sections.

### 2.2. Portable Vis/NIRS System and Spectra Acquisition

The portable Vis/NIR spectroscopy system was developed to collect spectra data. Figure 1 depicts a diagram of this system. Briefly, this portable Vis/NIR spectroscopy system was composed of a spectral data collection unit, a light source unit, and a computer. The spectral data collection unit consisted of a Vis/NIR spectrometer (AvaSpec-ULS2048CL-EVO-RS, Avantes Inc., Amsterdam, Netherlands), an optical probe, an optical fiber, and a USB 3.0 communication cable. The optical probe was a self-developed ring with 14 fibers in the middle, surrounded by hundreds of fibers, and a detection window of 30 mm. The spectral range of this Vis/NIR spectrometer was 200 to 1100 nm with a resolution of 0.06 nm, a sampling interval of 0.6 nm, and a signal-to-noise ratio (SNR) of 300:1, which ensured the superiority and reliability of the spectrometer. A tungsten halogen lamp with 20 W stable power was used as the light source.

Before spectra collection, the light source was preheated for nearly 30 min and the spectral information was then measured. The software used for collecting spectra was self-designed according to the software development kit (SDK) of the spectrometer supplier. The optimal parameters of spectra collection were set up using operation software, including a smoothness of 5 and an integral time of 100 ms. Each spectrum obtained was averaged using the 5 times spectra scans, which required only 0.5 s. The appropriate integration time was set to ensure the reflectivity of the standard whiteboard achieved approximately 80% of the maximum limit of detection. This Vis/NIR system was then adjusted using the reflection mode after the black and white corrections to decrease the adverse effects of the outside environment on the spectra’s accuracy. The samples to be tested were first placed on a flat table, then the optical detection probe was placed vertically on the sample surface for spectra acquisition. When collecting the spectra, the incoming light was perpendicular to the meat fiber direction for all muscle types. Finally, five different points on each sample were selected randomly to collect spectral data, and the average value was used as the final spectral reflectance of this sample.

### 2.3. pH Measurements

After collecting the spectra for each sample, the pH was immediately measured on samples using almost the same position where the spectral data were collected. To avoid pH changes caused by too long sample exposure, four portable digital pH meters (Testo 205, Testo A.G., Baden wurtenberg, Germany) were used to measure pH as soon as possible. The pH values were measured using traditional methods on the basis of industry standards. The probe of the portable digital pH meter was inserted directly into the meat about 1.5 cm deep from the meat surface after calibration with pH standard buffers. The values averaged from four different points were taken as the final measured pH values.

### 2.4. Color Measurements

The meat color parameters (lightness (*L**), redness (*a**), and yellowness (*b**)) were measured by a colorimeter (CM-600D, Konica Minolta Investment Ltd., Tokyo, Japan). This colorimeter was set to: the *L**, *a**, *b** system, illuminant to D65 (including ultraviolet), the standard colorimetric observer to a 2° visual angle, an aperture size of 8 mm, and measurement time of 1 s. Before measurements, the colorimeter needed to be calibrated using the white and black references. The colorimeter automatically measured four different points of each sample four times and output the average value, which avoids measurement errors. The average values were used as the final measured values.

### 2.5. Cooking Loss Measurements

To obtain samples with as uniform a size as possible to ensure the comparability of cooking loss values, the samples except for tenderloin cuts were first trimmed into cubes with a weight of about 100 g, then weighed to 0.01 g accurately. We placed the sample into a cooking bag and let the air out. Then, we placed the sample into a thermostat water bath (HH-4, Jintan ronghua Instrument Manufacturing Co., Ltd., Changzhou, China) and was cooked for 30 min at 70 °C. The boiled sample was cooled to 4 °C, then we wiped off the surface moisture and weighed the sample. The formula for cooking loss calculation is as follows:X = (m_1_ − m_2_)/m_1_ × 100%(1)
where X is the cooking loss of the sample (%), m_1_ is the weight of the tested sample before cooking (g), m_2_ is the weight of the tested sample after cooking (g).

### 2.6. Shear Force Measurements

The shear force value was measured to evaluate meat tenderness [25,26]. A CL-ML3 tenderness meter (Nanjing Mingao Instrument Equipment Co., Ltd., Nanjing, China) was used to measure the shear force according to the literature [27]. Briefly, the measurement range of this tenderness meter is 0 to 250 N, and the motor power for cutting samples is 20 W. Samples after cooking were cut into cubes (1 cm × 1 cm × 2 cm). The shear speed parameter was set to 5.0 mm/s for measurement. The shear force measurements were repeated at least 5 times and the average values were used as the final determination results.

### 2.7. Spectral Data Preprocessing

Spectral preprocessing methods were applied to remove the adverse effects of the external environment, sample density, and temperature variations on the spectral information and improve the model performance [28]. As the complexity of the spectra from different types of meat cut might negatively influence model performance, different spectral preprocessing methods, including multiplicative scatter correction (MSC) [29], standard normalized variate (SNV) [30], Detrend, Savitzky–Golay (S–G) smoothing [31], derivatives (1st Der and 2nd Der) [32] and their combinations were used for reducing the noise in the spectral information while preserving potential spectral data related to chemical information. The aim of the spectral preprocessing step was to develop a methodology that provides a systematic approach for further modeling analysis. Thus, different preprocessing combinations including S–G (5 points) + Detrend + MSC and S–G (5 points) + Detrend + SNV were also applied to improve the model performance. The above seven preprocessing methods were compared to determine which would effectively improve the model’s performance.

### 2.8. Multivariate Data Analysis

To analyze the differences in the physicochemical properties between the different cut types, one-way analysis of variance (ANOVA) was applied and Duncan’s multiple range tests were implemented in SPSS statistical software (version 19.0, SPSS Inc., Chicago, IL, USA). The statistical significance was defined as *p* < 0.05. Furthermore, to avoid the risk of the prediction models being constructed from non-informative or non-specific parts of the spectra and similar regression coefficients for different physicochemical properties, correlation analysis between the six physicochemical properties was carried out in SPSS. The statistical significance was defined as *p* < 0.05 (bilateral) and *p* < 0.01 (bilateral).

Partial least square regression (PLSR) is an algorithm for quantitative spectral decomposition commonly used to construct the fundamental relationship between independent variables (here, spectral data) and dependent variables (here, pH, *L**, *a**, *b**, cooking loss, and shear force values). PLSR can be used to build a linear model with numerous and strongly colinear variables, which is not only insensitive to collinear variables but also tolerant to large numbers of variables [33]. All spectral preprocessing and modeling procedures were performed by the algorithms developed in MATLAB mathematical software (R2014a, The Mathworks Inc., Natick, MA, USA).

The performance of the prediction models was evaluated on the basis of statistical assessment indicators: the determination coefficients of the calibration set (*R*_c_*^2^*) and prediction set (*R*_p_^2^), root mean square errors of the calibration set (RMSEC), and the prediction set (RMSEP), and the residual predictive deviation (RPD). Generally, a model with larger *R*_c_^2^ and *R*_p_^2^ values, and lower RMSEC and RMSEP values is more acceptable [34]. The RPD was calculated as the ratio of the standard deviation of the reference quality values in the prediction set to the RMSEP value. An RPD value lower than 2 indicates that the model is not recommended, a value between 2.0 and 2.4 indicates that the model can be used for rough screening, a value between 2.5 and 2.9 indicates the model can be used for screening, a value between 3.0 and 3.4 means the model can be used for quality control, between 3.5 and 3.9 indicates the model can be used for process control, and a value greater than 4.0 means that the model has excellent prediction performance and can be used for any application [35,36].

## 3. Results and Discussion

### 3.1. Quality Characteristics of Different Cut Types

Table 1 displays the variation range (from minimum to maximum), average values, standard deviation (SD), and coefficient of variation (CV) of pH, *L**, *a**, *b**, cooking loss, and shear force for different types of cut (silverside, back strap, oyster, fillet, thick flank, and tenderloin). Table 1 shows that the thick flank cut had the widest pH variation range, with the maximum (6.34) being obviously higher than other cuts. The fillet cut had a significantly higher pH (*p* < 0.05), which might be due to its greater more fat and fascial tissue contents. The pH ranges of the six cut types were close due to the pH differences between the different carcasses and between the different cut types were much smaller and more stable during the whole detection period [37].

The comparison results of lightness (*L**) showed that there were similar variation ranges between tenderloin, thick flank, and oyster cuts (*p* > 0.05), which were significantly higher than those of the silverside and back strap cuts (*p* < 0.05). The results revealed that the fillet cut had the highest lightness (*p* < 0.05), whereas the silverside and back strap cuts had the lowest lightness, which is consistent with their lean meat or fat and fascial tissue contents. From the *a** results in Table 1, the silverside, oyster, and fillet cuts had significantly higher redness (*p* < 0.05), and the silverside cut had the most balanced redness distribution because of having the largest amount of lean meat. From the comparison results of *b**, the fillet cut had the highest yellowness (*p* < 0.05) due to the large amounts of residual unbalanced distributed fat and connective tissues. The tenderloin cut had the lowest yellowness (*p* < 0.05) due to being all lean meat with the least fat.

The cooking loss results in Table 1 show that there were no significant differences between thick flank, oyster, fillet, and silverside cuts (*p* > 0.05). The tenderloin and back strap cuts had significantly higher WHCs (*p* < 0.05). From the shear force results, the tenderloin cut obviously had the lowest average of 38.80 N, indicating the highest tenderness, followed by oyster, then back strap, which is consistent with the trend in cooking loss. The higher the cooking loss value, the higher the tenderness, and the lower the shear force.

The correlations between the six physicochemical properties for all cuts of the carcass were calculated using their mean values and are displayed in Table 2. We found no strong correlations at the 0.01 level between these physicochemical properties except for *L** and *b** of 0.654, and *a** and *b** of 0.567, indicating that the risk of the prediction models being constructed from non-informative or non-specific parts of the spectra and similar regression coefficients for different physicochemical properties was avoided to a certain extent. To summarize, these data of physicochemical properties could be effectively used for subsequent modeling.

### 3.2. Sample Sets Division

To more intuitively compare the effect of biological variation on the prediction models, two sample set division methods were applied. For the sample set division based on the concentration gradient method, meat cuts from all the carcasses were arranged based on their measured quality attribute values in accordance with the spectral data, then the samples were divided into a calibration set and a prediction set. According to the 1/3 sample division principle, a randomly selected sample from every four samples was included in the prediction set for validating the model; whereas the rest were included in the calibration set for building the model. Table 3 provides the statistical results of sample set division based on the concentration gradient method. According to the usual 1/3 principle, the calibration set contained 112 samples and the prediction set contained 38 samples. For each quality parameter, the range of the reference measurements in the prediction set was covered by the range of the calibration set, which meant that the distributions of the reference data from the measurement samples in the calibration and prediction sets were almost equal, and bias in the distribution of the two sets was avoided.

For the sample set division based on carcasses, two carcasses per day for the first two days and one carcass per day for the next three days were used as the prediction set. All meat cuts from seven carcasses were included in the prediction set for validating the model, and the rest were included in the calibration set. The statistical results of sample set division based on carcasses are also shown in Table 3 to enable a more intuitive comparison. The calibration set contained 108 samples and the prediction set contained 42 samples. By contrast, the ranges of the *L**, *a**, and shear force of the prediction set were not covered by the range of the calibration set, which might have led to the instability of the prediction models.

### 3.3. Spectral Characteristics Analysis

Figure 2 shows the effects of the different spectral preprocessing methods. The wavelength region of the original spectra was 200–1100 nm. Due to more noise at both ends of the spectral curve, the data between 400 and 1000 nm were selected to develop the models for predicting physicochemical properties. The spectra of all the samples show the same change trend in Figure 2A. Furthermore, several obvious absorption peaks at 430, 550, 760, and 980 nm are positioned in the visible and near-infrared regions. Given the four meaningful wavelengths reported in previous studies by He et al. [38] and Cozzolino and Murray [39], the absorption peak at 430 nm in the visible region is related to the Soret absorption bond, which is attributed to traces of the erythrocytes of hemoglobin. The peak at 550 nm in the visible region is particularly related to the structure of meat myoglobin and oxyhemoglobin absorption. The absorption peak of 760 nm in the near-infrared region (700–1000 nm) is related to the absorption of deoxymyoglobin, mainly activated by the third overtone of O−H stretching. Another absorption peak occurred at about 980 nm, related to the 2nd overtone of the O−H stretching vibration appears, which is mainly related to water content. Because the shear force is a physicochemical index affected by many factors, it was difficult to find the characteristic absorption peak of a single corresponding group. Meat tenderness is mainly related to properties such as color and water holding capacity; therefore, the characteristic bands related to tenderness can be obtained indirectly by analyzing the absorption characteristics of the related factors affecting tenderness [40].

### 3.4. Spectral Preprocessing Analysis

Figure 2B–F shows the spectral curves after MSC, SNV, S–G smoothing, S–G smoothing + 1st Der, and S–G smoothing + 2nd Der preprocessing. The optimal spectral preprocessing method could not be determined only using the effects of the preprocessed spectra. The PLSR algorithm was employed to establish prediction models combined with the above different spectral pretreatments. The best spectral preprocessing for each quality parameter prediction was selected by comparing the model’s performance.

Table 4 provides the model results of the original spectra and the preprocessed spectral data after various pretreatments for pH, *L**, *a**, *b**, cooking loss, and shear force based on the 1/3 sample division. The bold indicates the best model. Overall, compared with the original spectra (no pretreatment), the predictive abilities for pH, *a**, *b**, and cooking loss were improved by several spectral preprocessing methods. MSC was performed to separate the scattering light signal from the spectrum and chemical absorption information and separate the direct reflection spectral data from the diffuse reflection data. MSC preprocessing reduced the scattering optical signal in the original reflection spectrum and the direct reflection signal, as shown in Figure 2B. The SNV was used to adjust the baseline drift and light scattering variation caused by the physicochemical structure of the samples. SNV preprocessing eliminated the effects of baseline variation resulting from the scattering optical signal, as shown in Figure 2C. Although there were differences in the physicochemical properties among the samples that were characterized by different spectra preprocessing methods to some extent, MSC produced a similar effect to SNV [41], which resulted in the similarity of model results for all quality parameters, as well as their preprocessing combinations (Table 4).

The modeling results of all quality parameters after S–G smoothing were similar to those using the original spectra (Table 4). The S–G smoothing eliminated the noise, enhanced the SNR of the spectra and did not considerably change the shape of the original spectra by calculating the mean values of the pixel gray level by the convolution algorithm, leading to *Y*-axis offset. However, if too few smoothing points are selected, the window is small and the effective information contained in the window is insufficient, which results in a poor filtering effect [42]. Thus, the spectra after five-point S–G smoothing, shown in Figure 2D, are similar to the original spectra in Figure 2A, which is why the modeling results were not effectively improved. Using the derivatives as the spectral preprocessing method would have introduced noise, so the derivatives were matched with S–G smoothing. We employed preprocessing combinations including S–G smoothing + 1st Der and S–G smoothing + 2nd Der (5 points).However, compared with the original spectra and other preprocessing methods, regardless of the quality attribute, the derivatives (1st Der and 2nd Der) preprocessing methods after S–G smoothing could not effectively improve the prediction ability of the models. After S–G smoothing + 1st Der preprocessing, several obvious spectral peaks were observed, as shown in Figure 2E. However, the derivative spectra might have weakened the effective quality information, although the first and the second derivatives eliminated the baseline shift independent of wavelength and linearly related to wavelength [43]. The poor prediction results of all physicochemical properties after S–G smoothing + 2nd Der preprocessing might have occurred because the second derivative spectrum produced some pseudo harmonic peaks and large amounts of noise, as shown in Figure 2F, which had little correlation with physicochemical properties. From Table 4, the single preprocessing method had a limited effect on improving the model, which might be due to the complexity of spectra collected from different cuts. The Detrend method could effectively eliminate the influence of the offset generated by the optical sensor when acquiring data on the later calculation. Deleting a trend from the data could effectively focus the analysis on the fluctuations of the data trend itself. In contrast, the most appropriate spectral preprocessing methods were determined for each quality attribute. The performance of the prediction models for pH, *L**, *a**, *b**, cooking loss, and shear force was improved by the preprocessing combinations S–G + Detrend + MSC and S–G + Detrend + SNV.

### 3.5. Establishment of Prediction Models Based on Combined Meat Cuts

Table 4 shows the prediction models of the quality parameters for combined meat cuts using the PLSR modeling method combined with different preprocessing methods. For pH, both MSC and SNV preprocessing effectively improved the performance of the prediction models. The number of optimal LVs of PLSR models was determined considering the minimum of the cross-validation standard error, which was implemented by leave-one-out cross-validation (LOOCV). The best determination coefficients of the calibration set (*R*_c_^2^) and prediction set (*R*_p_^2^) after S–G + Detrend + MSC preprocessing were 0.89 and 0.79, with RMSEC, RMSEP, and RPD of 0.03, 0.04, and 3.50, respectively, which are obviously better than those after MSC and SNV preprocessing. The modeling result based on carcass division is shown in Table 5, which shows that the best performance for pH prediction was poor, with an RPD value of less than two, which meant the application ability of the prediction model was unsatisfactory. Knight et al. [14] obtained ultimate pH (pH 24 h) predictions for the loin and topside muscle of Australian lamb with moderate *R*_p_^2^ values of 0.39 and 0.46, respectively. A PLS model result similar to our results was obtained: an *R*_p_ of 0.803 and RMSEP of 0.098 by Zhang et al. [44], who used Vis/NIR spectroscopy in the range of 350–1100 nm to predict pork pH. Balage et al. [10] also obtained similar PLSR model results for pH with an RPD of 2.1 using Vis/NIR spectroscopy, which indicated suitability for screening purposes when the longissimus dorsi samples from pigs were scanned intact.

As shown in Table 4, for lightness (*L**) prediction, the original spectra demonstrated satisfactory model performance with an RPD value of 2.20, but the model accuracy *R*_c_^2^ value was inferior to that after S–G + Detrend + SNV preprocessing, indicating it could be used for rough screening. The prediction ability after S–G + Detrend + SNV preprocessing achieved an *R*_c_^2^ as high as 0.85 with an RMSEC of 1.31, *R*_p_^2^ of 0.78, and an RMSEP of 1.80. Although S–G smoothing + 1st Der also showed the best calibration capability, its predictive result was poor with an RPD value of 1.74, which indicated that the model was not recommended. For the *L** prediction shown in Table 5, S–G + Detrend + SNV preprocessing obtained the best predictive ability with an RPD of 2.02, but this value is worse than that obtained based on the concentration gradient method.

For redness (*a**), the MSC and SNV effectively improved the calibration results but increased the number of LVs and their predictive performance was relatively poor. The huge differences between higher *R*_c_^2^ and lower *R*_p_^2^ values were indicative of overfitting, which would lead to poor performance in practical applications. The RPD values of the models ranged from 0.88 to 1.77, indicating that these models could not meet the recommended 2.0 threshold for rough screening [35]. The best model performance for *a** was obtained after S–G + Detrend + SNV preprocessing with an *R*_c_^2^ of 0.81 and RMSEC of 0.68, *R*_p_^2^ of 0.68, and RMSEP of 0.71. From Table 5, the optimal performance for *a** prediction was unsatisfactory due to the lower RPD values that were below two. After dividing sample sets based on carcasses, the number of LVs decreased, but the best model result after S–G + Detrend + MSC preprocessing was inferior to that obtained based on the concentration gradient method in Table 4.

For yellowness (*b**), the MSC and SNV provided extremely similar modeling results in terms of the values of *R*_c_^2^, RMSEC, *R*_p_^2^, and RMSEP. However, the MSC after S–G + Detrend preprocessing provided a more accurate model with a stronger predictive ability (RPD of 2.62 > 2.58). The optimal model performance for *b** prediction was an *R*_c_^2^ of 0.76, RMSEC of 0.63, *R*_p_^2^ of 0.75, RMSEP of 0.71, and RPD > 2. The model was found to be suitable for screening. Table 5 shows that the S–G + Detrend + MSC preprocessing method produced the best model performance, but the predictive ability was lower, as indicated by an RPD of 1.86. De Marchi et al. [45] obtained *a** and *b** predictions that were suitable for screening purposes and unreliable for *L** prediction on intact chicken breast using Vis/NIR spectroscopy in the range of 350–1800 nm. Balage et al. [10] obtained PLSR model results for *L**, *a**, and *b** of pig longissimus dorsi samples with RPD values of 2.3, 2.2, and 2.1, respectively, which indicated suitability for screening purposes.

For the cooking loss predictions in Table 4, the predictive performance was most effectively enhanced by S–G + Detrend + MSC and S–G + Detrend + SNV. The RPD value of each model ranged from 1.57 to 2.19 under different spectral preprocessing methods, probably because cooking loss is a complex attribute that does not have a direct connection to Vis/NIR spectra. Nevertheless, the MSC after S–G + Detrend preprocessing obtained a higher calibration accuracy and a lower LV of 4, indicating the larger reduction in the complexity of the model. Therefore, the best model for cooking loss prediction was obtained after S–G + Detrend + MSC preprocessing with an *R*_c_^2^ of 0.83, RMSEC of 1.85%, *R*_p_^2^ of 0.77, and RMSEP of 2.33%. From Table 5, cooking loss prediction based on carcass division was satisfactory, but the number of LVs was relatively higher, resulting in a complicated model. De Marchi et al. [6] obtained a worse PLSR model for cooking loss prediction with lower *R*_c_^2^ and RPD of 0.38 and 1.23, respectively, for intact beef samples; a worse model result for intact chicken samples was obtained with lower *R*_c_^2^ and RPD of 0.58 and 1.57, respectively, by De Marchi et al. [45].

For shear force, representing the tenderness, the satisfactory prediction models shown in Table 4 were obtained with high *R*_c_^2^ and *R*_p_^2^ values; and low RMSE values by applying MSC, SNV, and their combinations with S–G + Detrend. The most accurate prediction model of shear force had a higher *R*_c_^2^ of 0.84, lower RMSEC of 2.61 N, higher *R*_p_^2^ of 0.83, lower RMSEP of 2.64 N, and a moderate RPD value of 2.81 after S–G + Detrend + SNV preprocessing. The large tenderness differences among the different cut types benefitted these satisfactory results, which had a positive influence on the prediction models. The shear force prediction shown in Table 5 achieved good performance after S–G + Detrend + SNV preprocessing with an *R*_p_^2^ of 0.71 and an RPD value of 2.23, but slightly inferior performance to that based on the concentration gradient method. Huge differences in tenderness exist between different carcasses and between different cut types from one carcass. Sample sets division based on the concentration gradient method had a better effect on the tenderness distributions of the two sample sets. Previous reports showed the limited Vis/NIR spectroscopy ability in shear force prediction. Knight et al. [14] predicted shear force on day 5 of Australian lamb and obtained lower *R*^2^ values of 0.13 and 0.12 for the loin and topside muscle, respectively. Balage et al. [10] also obtained poor PLSR model results for WBSF with an *R*_c_^2^ of 0.48 and RPD of only 1.2, which means the model is not recommended for practical applications. De Marchi et al. [6] obtained unsatisfactory predictions of the WBSF for beef samples with an *R*_c_^2^ of 0.34 and RPD of 1.24, and prediction chicken samples with an *R*_cv_^2^ and an RPD of 1.20 [45]. Different cut types with differences in tenderness were used as experimental samples in the present study, thus enabling the satisfactory performance of the shear force [36].

To summarize, the prediction performances for all physicochemical parameters based on carcass division were inferior to those of dividing sample sets based on the concentration gradient method because the carcass division led to the uneven distribution of physicochemical parameters in the sample sets. If the ranges of physicochemical parameters in prediction sets are not covered in the calibration sets, the robustness of prediction models will be decreased. The comparisons based on two sample sets divisions proved that biological variation has a strong effect on the prediction models of physicochemical properties. The performance of the optimal PLSR models for each quality parameter is shown in Figure 3. The closer the sample points to the straight line, the more precise the predicted values. The PLSR model established by the original or the preprocessed spectra is presented as a vector in practical applications, which can be multiplied by the spectral vector to obtain the prediction results for different traits. In the follow-up application process, we can collect one spectral curve of the sample and substitute the data into the prediction models differently for different traits; then, the pH value, shear force, cooking loss, and other traits of this sample can be calculated to quantitatively detect the unknown sample quality index. Although Vis/NIR spectroscopy technology was previously studied for different quality attribute detections of livestock and poultry meats, such as beef, pork, mutton, chicken, and so on, the prediction model developed using only one type of meat cut still cannot be successfully applied to other types, which seriously limits the practical applicability of Vis/NIR spectroscopy technology. The results proved that Vis/NIR spectroscopy combined with chemometrics can be used to detect the physicochemical properties for various sheep meat cuts. The findings illustrate that Vis/NIR spectroscopy technology has the potential to simultaneously predict multiple physicochemical properties for different types of sheep meat cut.

## 4. Conclusions

In this work, we conducted a quantitative study of the physicochemical properties (pH, *L**, *a**, *b**, cooking loss, and shear force) of different types of commercial meat cut from sheep carcasses, including silverside, back strap, oyster, fillet, thick flank, and tenderloin cuts, based on Vis/NIR spectroscopy along with chemometric methods. The results indicated that the spectral preprocessing methods, to varying degrees, improved the robustness of the prediction models, and the MSC and SNV preprocessing combinations effectively achieved the most accurate prediction performances for different physicochemical properties. We conclude that Vis/NIR spectroscopy combined with the PLSR model and appropriate preprocessing methods has the ability to determine quality values for different commercial meat cuts in a rapid, nondestructive, and simultaneous prediction manner. In the absence of research on the quality evaluation of commercial cuts combined based on the Vis/NIRS technique, our findings provide a theoretical foundation and valuable information for the industrial application of Vis/NIR spectroscopy and this study will enhance its practical applicability. Although the models for each physicochemical property obtained relatively satisfactory prediction performance, further studies should be performed with a larger sample set (by adding more meat cut species or a larger sample number) to include a wider range of variation in the reference data for practical industrial implementation.

## Figures and Tables

**Figure 1 foods-10-01975-f001:**
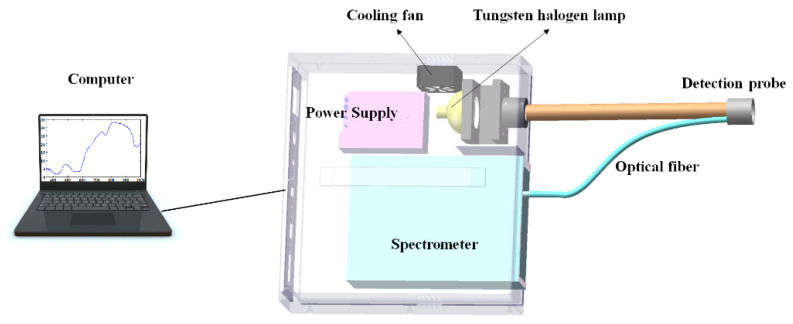
Schematic diagram of Vis/NIR spectroscopy system. This figure shows the experimental equipment for the quality detection of sheep meat cuts using Vis/NIR spectroscopy.

**Figure 2 foods-10-01975-f002:**
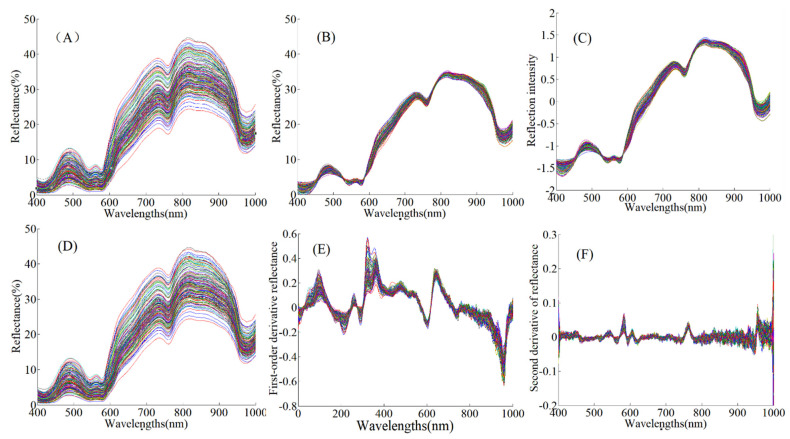
(**A**) Original spectra of all samples. (**B**) The spectra after MSC preprocessing. (**C**) The spectra after SNV preprocessing. (**D**) The spectra after S–G smoothing. (**E**) The spectra after S–G smoothing + 1st Der (5 points) preprocessing. (**F**) The spectra after S–G smoothing + 2nd Der (5 points) preprocessing.

**Figure 3 foods-10-01975-f003:**
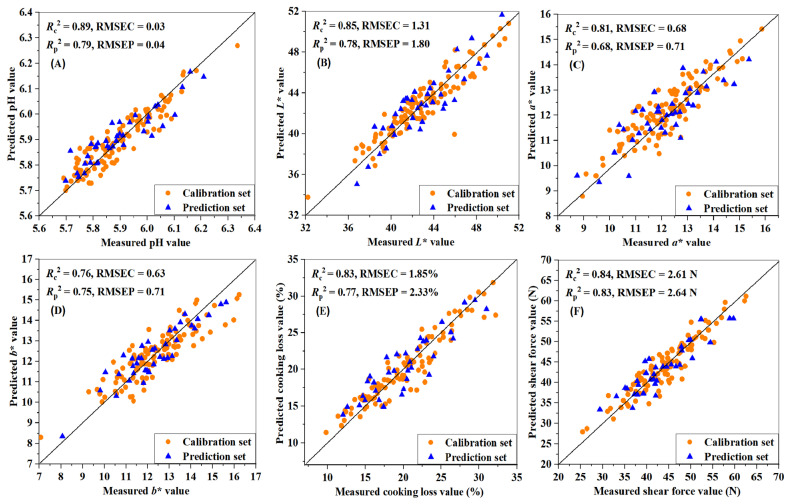
The prediction performances between the measured and predicted values for (**A**) pH, (**B**) *L**, (**C**) *a**, (**D**) *b**, (**E**) cooking loss and (**F**) shear force. The results show the ability to predict the physicochemical properties using Vis/NIR spectroscopy combined with the PLSR algorithm. The closer the sample points to the straight line, the more precise the predicted value, which illustrates that the prediction model is more accurate and robust.

**Table 1 foods-10-01975-t001:** Descriptive statistics of the physicochemical properties characteristics for different meat cut types.

Properties	Cut Type	Maximum	Minimum	Mean	SD	CV (%)
pH	tenderloin	6.10	5.74	5.86 ^c^	0.09	1.53
	thick flank	6.34	5.75	5.94 ^b^	0.14	2.36
	oyster	6.09	5.81	5.94 ^b^	0.08	1.35
	fillet	6.18	5.90	6.04 ^a^	0.08	1.32
	silverside	6.04	5.69	5.79 ^d^	0.08	1.38
	back strap	6.08	5.70	5.82 ^cd^	0.09	1.55
	all cuts	6.34	5.69	5.90	0.12	2.03
*L**	tenderloin	47.64	38.45	43.19 ^b^	2.46	5.70
	thick flank	48.80	39.87	42.76 ^b^	2.05	4.79
	oyster	47.44	39.26	43.19 ^b^	1.90	4.40
	fillet	51.03	42.21	47.43 ^a^	2.30	4.85
	silverside	42.73	36.61	39.00 ^d^	1.72	4.41
	back strap	44.70	32.21	41.55 ^c^	2.44	5.87
	all cuts	51.03	32.21	42.85	3.29	7.68
*a**	tenderloin	14.48	8.95	11.39 ^c^	1.36	11.94
	thick flank	14.15	10.36	11.97 ^bc^	1.06	8.86
	oyster	14.91	10.36	12.49 ^ab^	1.17	9.37
	fillet	15.86	9.59	12.39 ^ab^	1.53	12.35
	silverside	15.03	11.27	13.06 ^a^	0.90	6.89
	back strap	14.66	8.74	11.91 ^bc^	1.43	12.01
	all cuts	15.86	8.74	12.20	1.35	11.07
*b**	tenderloin	13.54	8.07	10.90 ^d^	1.17	10.73
	thick flank	14.53	10.05	12.40 ^bc^	1.00	8.06
	oyster	15.09	9.79	12.82 ^b^	1.11	8.66
	fillet	16.39	10.67	13.95 ^a^	1.58	11.33
	silverside	14.04	10.51	11.79 ^c^	0.96	8.14
	back strap	14.74	7.07	12.09 ^c^	1.34	11.08
	all cuts	16.39	7.07	12.33	1.52	12.33
Cooking loss (%)	tenderloin	29.46	15.87	22.63 ^a^	3.29	14.54
	thick flank	30.67	11.84	18.99 ^b^	4.81	25.33
	oyster	28.07	9.81	17.67 ^b^	4.42	25.01
	fillet	32.20	12.31	19.93 ^b^	5.27	26.44
	silverside	28.80	12.04	18.46 ^b^	4.59	24.86
	back strap	31.87	16.83	23.18 ^a^	4.12	17.77
	all cuts	32.20	9.81	20.13	4.85	24.09
Shear force (N)	tenderloin	49.55	29.41	38.80 ^c^	4.80	12.37
	thick flank	57.67	37.56	44.49 ^b^	5.19	11.67
	oyster	50.26	34.53	42.31 ^bc^	4.27	10.09
	fillet	62.59	44.25	49.87 ^a^	4.98	9.99
	silverside	54.05	35.85	44.30 ^b^	4.92	11.11
	back strap	62.26	25.46	42.95 ^b^	9.97	23.21
	all cuts	62.59	25.46	43.90	6.93	15.79

*L**, *a**, and *b** represent lightness, redness, and yellowness, respectively; SD, standard deviation; CV, coefficient of variation; a~d indicate significant differences between different meat cut types (*p* < 0.05); the same letter represents no significant difference (*p* > 0.05).

**Table 2 foods-10-01975-t002:** Correlations between the six physicochemical properties of the combined meat cut types.

Properties	pH	*L**	*a**	*b**	Cooking Loss	Shear Force
pH	1					
*L* ^*^	0.406 **	1				
*a* ^*^	−0.188 *	−0.074	1			
*b* ^*^	0.224 **	0.654 **	0.567 **	1		
Cooking loss	−0.224 **	−0.169 *	0.017	−0.245 **	1	
Shear force	0.266 **	0.139	−0.052	0.152	−0.127	1

** The correlation was significant at 0.01 level (bilateral). * The correlation was significant at 0.05 level (bilateral).

**Table 3 foods-10-01975-t003:** Reference measurement of quality attributes in the calibration and prediction sets based on two sample set divisions.

Properties	Subsets	Number	Range	Mean	SD	CV (%)
pH	Calibration	112	5.69–6.34	5.90	0.13	2.20
	Prediction	38	5.70–6.16	5.90	0.14	2.37
*L**	Calibration	112	32.21–51.03	42.83	3.35	7.82
	Prediction	38	36.80–50.40	42.92	4.11	9.58
*a**	Calibration	112	8.74–15.86	12.21	1.36	11.14
	Prediction	38	9.09–15.37	12.26	1.75	14.27
*b**	Calibration	112	7.07–16.39	12.33	1.56	12.65
	Prediction	38	9.28–15.98	12.31	1.86	15.11
Cooking loss (%)	Calibration	112	9.81–32.20	20.11	4.92	24.47
	Prediction	38	12.04–30.97	20.20	5.10	25.25
Shear force (N)	Calibration	112	25.46–62.59	43.95	7.06	16.06
	Prediction	38	29.41–58.88	43.76	7.41	16.93
pH	Calibration	108	5.69–6.33	5.91	0.13	2.20
	Prediction	42	5.70–6.21	5.87	0.11	1.87
*L**	Calibration	108	32.21–50.67	42.80	3.07	7.17
	Prediction	42	36.74–51.03	42.98	3.85	8.96
*a**	Calibration	108	8.74–15.37	12.06	1.34	11.11
	Prediction	42	9.09–15.86	12.57	1.31	10.42
*b**	Calibration	108	7.07–16.39	12.24	1.51	12.34
	Prediction	42	9.28–16.13	12.54	1.54	12.28
Cooking loss (%)	Calibration	108	9.81–32.20	20.86	5.24	25.12
	Prediction	42	12.31–26.59	18.48	3.32	17.97
Shear force (N)	Calibration	108	26.46–62.59	43.43	7.02	16.16
	Prediction	42	25.46–62.26	45.00	6.66	14.80

*L**, *a**, and *b** represent lightness, redness, and yellowness, respectively; SD, standard deviation; CV, coefficient of variation.

**Table 4 foods-10-01975-t004:** The comparison result of PLSR models established based on the concentration gradient method for the combined meat cuts.

Properties	Preprocessing Methods	Number of LVs	Calibration		Prediction		RPD
			*R* _c_ ^2^	RMSEC	*R* _p_ ^2^	RMSEP	
pH	Original spectra	10	0.36	0.12	0.19	0.22	0.64
	MSC	9	0.82	0.04	0.69	0.06	2.33
	SNV	9	0.82	0.04	0.69	0.06	2.33
	S–G smoothing	12	0.45	0.11	0.22	0.20	0.70
	S–G + 1st Der	9	0.48	0.10	0.42	0.32	0.44
	S–G + 2nd Der	10	0.50	0.10	0.64	0.07	2.00
	**S–G + Detrend + MSC**	**9**	**0.89**	**0.03**	**0.79**	**0.04**	**3.50**
	S–G + Detrend + SNV	8	0.89	0.03	0.77	0.05	2.80
*L**	Original spectra	11	0.71	1.82	0.69	1.87	2.20
	MSC	11	0.77	1.61	0.69	1.94	2.12
	SNV	11	0.77	1.61	0.71	1.87	2.20
	S–G smoothing	11	0.74	1.70	0.67	1.89	2.17
	S–G + 1st Der	9	0.85	1.30	0.61	2.36	1.74
	S–G + 2nd Der	8	0.68	1.79	0.57	2.92	1.41
	S–G + Detrend + MSC	7	0.85	1.31	0.76	1.82	2.26
	**S–G + Detrend + SNV**	**7**	**0.85**	**1.31**	**0.78**	**1.80**	**2.28**
*a**	Original spectra	10	0.65	0.81	0.46	1.00	1.75
	MSC	13	0.67	0.78	0.30	0.99	1.77
	SNV	12	0.67	0.78	0.30	1.25	1.40
	S–G smoothing	10	0.62	0.85	0.43	1.35	1.30
	S–G + 1st Der	9	0.71	0.73	0.46	1.34	1.31
	S–G + 2nd Der	5	0.27	1.20	0.24	1.98	0.88
	S–G + Detrend + MSC	8	0.80	0.68	0.66	0.72	2.43
	**S–G + Detrend + SNV**	**9**	**0.81**	**0.68**	**0.68**	**0.71**	**2.46**
*b**	Original spectra	7	0.58	1.02	0.52	1.46	1.27
	MSC	9	0.63	0.89	0.62	1.00	1.86
	SNV	10	0.64	0.90	0.62	0.99	1.88
	S–G smoothing	8	0.61	0.93	0.69	0.88	2.11
	S–G + 1st Der	5	0.56	1.04	0.50	1.13	1.65
	S–G + 2nd Der	7	0.57	1.03	0.52	1.52	1.22
	**S–G + Detrend + MSC**	**7**	**0.76**	**0.63**	**0.75**	**0.71**	**2.62**
	S–G + Detrend + SNV	7	0.75	0.64	0.75	0.72	2.58
Cooking loss	Original spectra	10	0.71	2.71	0.53	3.09	1.65
	MSC	8	0.82	1.83	0.62	2.90	1.76
	SNV	8	0.82	1.82	0.62	2.90	1.76
	S–G smoothing	12	0.69	2.73	0.59	3.10	1.65
	S–G + 1st Der	7	0.88	1.62	0.66	2.83	1.80
	S–G + 2nd Der	6	0.68	2.77	0.52	3.25	1.57
	**S–G + Detrend + MSC**	**4**	**0.83**	**1.85**	**0.77**	**2.33**	**2.19**
	S–G + Detrend + SNV	4	0.83	1.78	0.76	2.34	2.18
Shear force	Original spectra	10	0.74	3.44	0.42	5.89	1.26
	MSC	10	0.88	2.36	0.74	3.41	2.17
	SNV	10	0.88	2.36	0.73	3.41	2.17
	S–G smoothing	9	0.76	3.38	0.66	3.91	1.90
	S–G + 1st Der	5	0.62	4.18	0.47	5.56	1.33
	S–G + 2nd Der	8	0.74	3.43	0.38	6.34	1.17
	S–G + Detrend + MSC	7	0.82	2.62	0.81	2.67	2.78
	**S–G + Detrend + SNV**	**8**	**0.84**	**2.61**	**0.83**	**2.64**	**2.81**

*R*_c_^2^, coefficient of determination of calibration set; RMSEC, the root mean squared error of calibration set; *R*_p_^2^, coefficient of determination of prediction set; RMSEP, the root mean squared error of prediction set; LVs, latent variables; 1st Der, first derivative; 2nd Der, second derivative; MSC, multiplicative scatter correction; SNV, standard normalized variate; S–G, Savitzky–Golay smoothing; RPD, residual predictive deviation.

**Table 5 foods-10-01975-t005:** The comparison result of PLSR models established based on carcass division for the combined meat cuts.

Properties	Preprocessing Methods	Number of LVs	Calibration		Prediction		RPD
			*R* _c_ ^2^	RMSEC	*R* _p_ ^2^	RMSEP	
pH	S–G + Detrend + MSC	7	0.74	0.05	0.69	0.06	1.83
*L**	S–G + 2nd Der	5	0.78	1.60	0.65	1.91	2.02
*a**	S–G + Detrend + MSC	6	0.72	0.58	0.63	0.81	1.62
*b**	S–G + Detrend + MSC	8	0.70	0.81	0.71	0.83	1.86
Cooking loss	S–G + Detrend + MSC	10	0.82	2.05	0.75	1.69	1.96
Shear force	S–G + Detrend + SNV	8	0.89	2.23	0.71	2.98	2.23

*R*_c_^2^, coefficient of determination of calibration set; RMSEC, the root mean squared error of calibration set; *R*_p_^2^, coefficient of determination of prediction set; RMSEP, the root mean squared error of prediction set; LVs, latent variables; 2nd Der, second derivative; MSC, multiplicative scatter correction; SNV, standard normalized variate; S–G, Savitzky–Golay smoothing; RPD, residual predictive deviation.

## Data Availability

The data presented in this study are available on request from the corresponding author.

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
