# Peer review of "Rapid Nondestructive Simultaneous Detection for Physicochemical Properties of Different Types of Sheep Meat Cut Using Portable Vis/NIR Reflectance Spectroscopy System"

_foods, 2021, doi:10.3390/foods10091975_

Round 1

Reviewer 1 Report

The title describes a study of physical properties. However, pH is not a physical property but a chemical property.

Lines 14-15: „The visible and near-infrared spectroscopy (Vis/NIRS) models for meat quality evaluation using only one type of meat cut are not be suitable for other types”

The statement is incorrect. For beef and pork, the NIR methods used are based on several types of meat

Lines 20-21: Rp2 and RPD were replaced for cooking loss and shear force

Chapter 2.2. What was the layer thickness of the sample? From the average of how many sub-spectra did the system generate the spectra?

Lines 120-121 : “The spectral range of this Vis/NIR spectrometer was 200 to 1100 nm with a resolution of 0.06 nm, sampling interval of 0.6 nm…”

The visible range starts at 380 nm. Based on the data given, does this instrument measure in UV? I do not think so.

Lines 304-305: The calibration set contained 112 samples and the prediction set contained 38 samples.” Same here, lines 310-311: “The calibration set contained 108 samples and the prediction set contained 42 samples. “ This part needs to be reworded because the different sample numbers are not clear at first reading. Why was the ratio 112:38 or 108:42 chosen instead of the usual 2/3:1/3?

Lines 321-322: Why not use the measuring range of the instrument between 1000-1100 nm? A lot of valuable information was lost.

Lines 332-333: “Another absorption peak occurred at about 980 nm, related to O-H third stretch  overtone, which is mainly related to water content.” The statement is not correct. At 980 nm the 2nd overtone of the OH stretching vibration appears.

Line 386 Table 4: Number of LVs - I guess this is the number of latent variables. These values are too high for the sample size. With this sample number, we are working with a maximum of 10 latent variables, otherwise the function is overfitted. But even so, the validation R2 of the functions and the RPD are still quite weak. I have the same conclusion for the data in Table 5.

The data in Tables 3, 4 and 5 should be rearranged to show the range of the classical measurement data alongside the RMSEP values of the NIR calibration, because only then can the "goodness of fit" of the calibration be compared.

Fig 3: The data in Figure 3 also support my earlier view that the calibrations are over-fitted. The R2 of the calibration and the validation cannot be the same (pH), but they cannot be too different (a*, WHC%). The same is confirmed by the fact that the difference between RMSEC and RMSEP is too large for a* and WHC.

The conclusion that NIR+ chemometric data analysis is suitable for meat is not new.

The evaluation data reported in the manuscript are poor and the calibrations are over-fitted.

A full review and new calibration contexts are needed.

Reviewer 2 Report

Dear Authors,

The paper presented for review concerns rapid nondestructive detection of physical properties of sheep meat cut using portable Vis / NIR system. The manuscript was written in a very clear, coherent and understandable language. The conducted research and their results are very interesting and promising from the perspective of using this technique in industrial conditions. It may be necessary in future to carry out studies on a larger number of animals, but I suppose that, the presented studies are preliminary studies.

 Some detailed comments:

 - Line 58-61, please replace "different parts of meat" with: "different cuts of the carcass" and add: have different tissues composition instead of "have different connective tissues",

- is the small-tailed Han sheep breed popular in China? please add short information in material methods section (Line 95-96)

- was the internal temperature of the meat sample measured (controlled) during the cooking-loss determination or only the heating time (30 minutes), please explain

- Line 292, "the correlation between the six physical properties were calculated ..." but what means the value of these six properties? Is this the mean value for all parts of the carcass? please explain.

 - in the tables there are "a -d indicate significant differences between meat cut types", why there are only four letters (a, b, c, d), since there were six meat cut types? Please explain.

- Please add in the footnote of the tables explanation, what do the abbreviations MSC, SNV, S-G etc. mean, this also applies to the text of the work.

 Best regards

Round 2

Reviewer 1 Report

The authors have corrected the errors listed and made new calculations.
I accept the manuscript in this form